# Resolving Entangled *J*_H-H_-Coupling Patterns for Steroidal Structure Determinations by NMR Spectroscopy

**DOI:** 10.3390/molecules26092643

**Published:** 2021-04-30

**Authors:** Danni Wu, Kathleen Joyce Carillo, Jiun-Jie Shie, Steve S.-F. Yu, Der-Lii M. Tzou

**Affiliations:** 1Institute of Chemistry, Academia Sinica, Nankang, Taipei 11529, Taiwan; tomo20170414@gmail.com (D.W.); kjcarillo@yahoo.com (K.J.C.); shiejj@gate.sinica.edu.tw (J.-J.S.); Sfyu@gate.sinica.edu.tw (S.S.-F.Y.); 2International Graduate Program, SCST, Academia Sinica, Nankang, Taipei 11529, Taiwan; 3The Department of Applied Chemistry, National Yang Ming Chiao Tung University, Hsinchu 30013, Taiwan; 4Department of Applied Chemistry, National Chia-Yi University, Chia-Yi 60004, Taiwan

**Keywords:** steroid hormones, proton chemical shifts, *J* scalar coupling constants, structural determinations, fingerprint patterns

## Abstract

For decades, high-resolution ^1^H NMR spectroscopy has been routinely utilized to analyze both naturally occurring steroid hormones and synthetic steroids, which play important roles in regulating physiological functions in humans. Because the ^1^H signals are inevitably superimposed and entangled with various *J*_H–H_ splitting patterns, such that the individual ^1^H chemical shift and associated *J*_H–H_ coupling identities are hardly resolved. Given this, applications of thess information for elucidating steroidal molecular structures and steroid/ligand interactions at the atomic level were largely restricted. To overcome, we devoted to unraveling the entangled *J*_H–H_ splitting patterns of two similar steroidal compounds having fully unsaturated protons, i.e., androstanolone and epiandrosterone (denoted as **1** and **2**, respectively), in which only hydroxyl and ketone substituents attached to C3 and C17 were interchanged. Here we demonstrated that the *J*_H–H_ values deduced from **1** and **2** are universal and applicable to other steroids, such as testosterone, 3β, 21-dihydroxygregna-5-en-20-one, prednisolone, and estradiol. On the other hand, the ^1^H chemical shifts may deviate substantially from sample to sample. In this communication, we propose a simple but novel scheme for resolving the complicate *J*_H–H_ splitting patterns and ^1^H chemical shifts, aiming for steroidal structure determinations.

## 1. Introduction

It is well accepted that one-dimensional (1D) ^1^H NMR spectroscopy serves as the most popular and quantitative analytical tool for small molecules [1] as well as for metabolomics [2,3,4]. The resulting ^1^H NMR spectra reveal distinct ^1^H spectral patterns for identifying the molecular structure and yet very often they are superimposed and coupled with neighboring spins via spin–spin scalar interactions. Typically, in steroidal compounds, resulting from the spin-spin couplings among adjacent ^1^H nuclei, the ^1^H signals are superimposed on each other and thereby convoluted severely with various scalar (*J*_H–H_; spin–spin) coupling patterns. As an outcome, it is extremely difficult to identify each of their ^1^H resonances as well as the corresponding *J*_H–H_ coupling constants. Although the ^1^H NMR spectroscopy acts as an important tool in analyzing molecular structure and dynamics at atomic resolution, and yet its applications to steroidal compounds are still limited.

Regardless of structural similarity, steroid hormones mediate a variety of physiological functions in humans [5,6,7,8,9]. In the steroid hormones, their ^1^H spectra are actually quite different from one another. Each ^1^H nucleus exists in a unique but slightly different local environment such that the associated scalar coupling pattern is made of either a doublet or a multiple of doublets. For those ^1^H signals that are partially overlapped or superimposed, one has to make additional efforts and tedious measurements to resolve. Conventionally, one needs to apply traditional two-dimensional (2D) heteronuclear correlation spectroscopy, such as heteronuclear single-quantum correlation (HSQC) [10], heteronuclear multiple bond correlation (HMBC) [11], and homonuclear correlation spectroscopy (COSY) [12] experiments, to verify each of the ^1^H chemical shifts and to unravel associated *J*_H–H_ coupling constants. Aside from the traditional 2D NMR experiments, 2D *J*_H–H_-resolved spectroscopy is available for better resolving the *J*_H–H_ splitting patterns in various schemes, for details see review article [13] and references therein. These *J*_H–H_-resolved 2D methods are aimed at detecting *J*_H–H_ at the expense of signal sensitivity, which requires more measurement time than traditional 2D NMR experiments. Typically, a high field spectrometer (>600 MHz) is recommended to achieve higher spectral resolution. For interpretation of ^1^H NMR spectra of complex compounds, in terms of their ^1^H chemical shifts and *J*_H–H_ coupling constants, an iterative ^1^H Full Spin Analysis (HiFSA) method was first developed by Pauli and coworkers based on quantum mechanical calculations [14,15,16]. So far, there are no other alternatives that one can utilize to resolve ^1^H chemical shifts and *J*_H–H_ coupling constants, free from field strength-dependent measurements and/or laborious processes.

In this work, we initiate a new approach that allows one to uncover the superimposed ^1^H chemical shifts as well as the *J*_H–H_ coupling constants straightforwardly and reliably, aiming for structure determination.

## 2. Results

### 2.1. Chemical Shift Assignments

In the NMR spectra of steroidal compounds, the intrinsic *J*_H–H_ coupling constants are basically invariant whereas their ^1^H chemical shifts often vary substantially. For proof-of-concept, we selected two nearly identical steroidal compounds, i.e., compounds **1** and **2**, and measured their ^1^H spectra for comparison. Molecular structures of **1** and **2** do show the same steroidal scaffold with slight modifications, in which the substituents attached to C3 and C17, i.e., ketone and hydroxyl groups, are interchanged (Scheme 1).

As demonstrated by the spectrum of **1** (Figure 1a), several ^1^H signals are superimposed and entangled with multiple *J*_H–H_ splitting patterns, in the ranges of δ = 0.9–1.0, 1.05–1.09, 1.33–1.5, 1.55–1.65, and 1.94–2.03 ppm. Same for **2** (Figure 1c), more than half of the ^1^H signals are seriously overlapped with multiple *J*_H–H_ splitting patterns, in the ranges of δ = 1.0–1.1, 1.15–1.47, 1.53–1.72, and 1.73–1.80 ppm. Regardless of the structural similarity, the two spectra are so dissimilar and complicate that are difficult to assign.

Figure 1. Solution NMR analysis of the steroidal compounds 1 and 2. 1D ^1^H spectrum of (a) 1 and (c) 2. Note that 1 and 2 share a high degree of structural similarity (see Scheme 1), and yet their ^1^H NMR spectra are rather dissimilar. In (a), half of the ^1^H signals are superimposed and feature multiple *J*_H–H_ splitting patterns, in the ranges of δ = 0.9–1.0, 1.05–1.09, 1.33–1.5, 1.55–1.65, and 1.94–2.03 ppm. Whereas, in (c), more than half of the ^1^H signals are overlapped and coupled with multiple *J*_H–H_ splitting patterns, in the ranges of δ = 1.0–1.1, 1.15–1.47, and 1.53–1.80 ppm. Given signal overlapping and the signal scrambling effect, the two spectral patterns are dissimilar. The ^1^H chemical shift assignments were determined from the 2D HSQC experiment accordingly (Appendix A). (b,d) are a simulated spectrum of 1 and 2, based on the ^1^H chemical shift assignments and associated *J*_H–H_ values (Table 1), using the Daisy software program (version 2.0.0 Bruker BioSpin GmbH, Rheinstetten, Germany); for details see text. For better clarity, these spectra are present on different scales.

To facilitate the chemical shift assignments, we then conducted 2D ^1^H/^13^C HSQC and DEPT experiments to identify the ^1^H and ^13^C chemical shifts of **1** and **2** (see Appendix A), respectively. The ^1^H chemical shifts were further confirmed by 2D HMBC and COSY (correlation spectroscopy) experiments (Appendix A). Both the ^1^H and ^13^C chemical shift assignments were determined (Table 1 and Appendix A). To confirm the spectral dissimilarity is mainly due to ^1^H chemical shift differences but not associated *J*_H–H_ coupling constant variations, we also carried out an iterative simulation analysis to unravel the *J*_H–H_ values of **1** and **2** (Table 1), respectively, to be elaborated below. As suggested, most of their ^1^H chemical shifts (18 out of 24) differ by 0.1–1.0 ppm, equivalent to 50–500 Hz as measured by a 500 MHz spectrometer. In contrast, most of the *J*_H–H_ values remained basically invariant. There are only a few residues in rings *A* and *D*, indicating deviations in the range of 0.3–5.7 Hz. Thus, it was confirmed that the spectral dissimilarity between **1** and **2** is primarily due to the ^1^H chemical shift deviations, but not the variations of the *J*_H–H_ coupling constants.

### 2.2. Resonance Hopping Effect

Note that the ^1^H resonances of **1** and **2** display different sequential ordering from the downfield to upfield regions. In the case of **1**, as revealed by steroidal ring *B*, the signals arising from H7β (1.730 ppm), H5 (1.526 ppm), H8 (1.486 ppm), H6β (1.358 ppm), H6α (1.350 ppm), H7α (0.934 ppm) and H9 (0.766 ppm) displayed in descending order. Whereas in **2**, the same signals followed a dissimilar sequential ordering, i.e., H7β (1.847 ppm), H8 (1.628 ppm), H6β (1.359 ppm), H6α (1.358 ppm), H5 (1.173 ppm), H7α (1.046 ppm), and H9 (0.750 ppm). We termed this as a resonance hopping effect or sequential disordering effect. As suggested, a modification of certain substituents, in this case, an interexchange of ketone and hydroxyl groups, could lead to a substantial change of the chemical environment in remote and non-related sites. For example, the hopping effect was also detected from those signals arising from steroidal ring *C* as well, i.e., H12β (1.848 ppm in **1** and 1.759 ppm in **2**), H15α (1.593 ppm in **1** and 1.961 ppm in **2**), H12α (1.063 ppm in **1** and 1.235 ppm in **2**), and H14 (0.982 ppm in **1** and 1.330 ppm in **2**). Notice that the ^1^H chemical shifts of **1** and **2** vary in a wide range from −0.62 to +0.107 ppm, not limited to rings *A* and *D*. Given this hopping effect, aside from the signal superimpositions, the signal assignments of **1** and **2** are rather complicated.

### 2.3. Fingerprint Pattern Identification

In steroidal compounds, each ^1^H nucleus is experienced in a unique chemical environment, in which the respective scalar coupling pattern is made of either a doublet or a multiple of doublets, depending on the adjacent ^1^H nuclei. As reported in the literature [17,18], each ^1^H signal typically is involved three to four spin-spin couplings of different “sizes”. And the overall splitting pattern is mainly dependent on the number of the *J*_H–H_ coupling constants and their sizes. As referring to the size variations in a unit of Hz, the geminal coupling constants (^2^*J*_gem_) distributed in the range of −12 to −14 Hz, the axial-axial coupling constants (^3^*J*_ax–ax_) vary within 10.5 to 14.5 Hz, the axial-equatorial coupling constants (^3^*J*_ax–eq_) within 3.5 to 5.0 Hz, the equatorial-equatorial coupling constants (^3^*J*_eq–eq_) near 3.0 Hz, as well as other (^4^*J*_H–H_) coupling constants below 3–4 Hz. To facilitate *J*_H–H_ coupling constant determinations, we here divided the *J*_H–H_ values of steroids into four categories based on their “sizes”, namely large (14–10 Hz), medium-large (9–6 Hz), medium (5–3 Hz) and small (below 3 Hz). As revealed in Figure 2, various scenarios, including “*doublet*” of “*quartets*”, “*triplet*” of “*doublets*”, and “*triplet*” of “*quartets*”, are representative of the *J*_H–H_ values of certain combinations. In the light of the distinct spectral patterns, one can identify the associated *J*_H–H_ coupling constants directly from its 1D spectrum. For example, the H1β signal of **1** (Figure 1a) resembles that of “*doublet*” of “*quartets*” pattern (Figure 2b), and the H12α signal of **2** (Figure 1c) is quite similar to that of “*triplet*” of “*doublets*” feature (Figure 2e).

For those ^1^H signal patterns free from any signal overlap, one can follow the abovementioned simulation analysis to deduce the *J*_H–H_ values directly. For those signals with partial or serious overlap, one needs to pay extra efforts to verify manually possible scenarios in different combinations of chemical shifts and *J*_H–H_ values, which is considered labor-intensive and time-consuming. For this, we proposed an iterative simulation scheme for unraveling the ^1^H chemical shits and the *J*_H–H_ values stepwise, to be elaborated below.

### 2.4. Resolving Entangled J_H–H_ Coupling Patterns

Because of the spectral complexity, it is highly advisable to resolve the *J*_H–H_ coupling constants stepwise, first starting from those signals free from signal overlap, then moving on to those with partial signal overlap, and finally to those with serious signal overlap. As suggested, one shall first eliminate impossible combinations and then worked on the simulation for better fitting of the experimental spectrum. Having the chemical shifts deduced from the 2D HSQC spectra, we then uncovered the entangled *J*_H–H_ splitting patterns iteratively. In the 1D spectra of **1** and **2** (Figure 1a,c), the most difficult part arose from those ^1^H signals with serious overlap, i.e., H8/H16α/H11β in **1** as well as H11β/H6β/H6α/H14/H4β in **2**. In these cases, we mainly focused on the uncovered upfield portion for simulation to extract the *J*_H–H_ coupling constants. With our great efforts, we achieved to determine the *J*_H–H_ coupling constants of **1** and **2**, respectively. As indicated, the simulated ^1^H spectra of **1** and **2** (Figure 1b,d) are in good agreement with the experimental results (Figure 1a,c), indicating that the *J*_H–H_ values deduced from **1** and **2** are validated.

To validate whether the *J*_H–H_ values deduced from **1** and **2** apply to other steroidal compounds, we chose four steroidal compounds, i.e., testosterone (**3**), 3β, 21-dihydroxypregna-5-en-20-one (**4**), prednisolone (**5**), and estradiol (**6**) (Scheme 1) for NMR spectral analysis. According to the *J*_H–H_ values deduced from **1** and **2** (Table 1) and the ^1^H chemical shifts determined from their 2D HSQC spectrum (Appendix A), we simulated the ^1^H spectra and divided it into four subspectra, corresponding to rings *A* (H1-H5), *B* (H6-H9), *C* (H11-H12), and *D* (H14-H17), respectively, to reduce the spectral overlaps. As displayed (Figure 3 and Appendix A), the ^1^H signals were better resolved such that one could identify characteristic spectral patterns. For example, similar spectral patterns were found in **1** and **3** (Figure 1a and Figure 3), i.e., H7α, H11α, H11β, H12β, H14, H15β, H16β, and H17α; in **2** and **6** (Figure 2c and Appendix A), i.e., H3β, H7β, H9, H11α, and H12α; in **4** and **5** (Appendix A), i.e., H1, H2, H4, H6β, and H7α, and in **3** and **5** (Figure 3 and Appendix A), i.e., H6α and H7α. For more applications, we then deduced several *J*_H–H_ values from different ring structures (Table 2), in association with human steroid hormones [19].

Here we proposed a simple and iterative scheme for unraveling *J*_H–H_ values, aiming for molecular structure determination by solution NMR spectroscopy (Scheme 2). In this scheme, we first measured 1D ^1^H and 2D ^1^H/^13^C HSQC spectra of the steroidal compound to be studied. For un-overlapped signals, both the chemical shifts and *J*_H–H_ values can be determined straightforward, as described above. On the other hand, for each of the overlapped signals, one can identify how many ^1^H signals are needed for database searching. Given their ^1^H chemical shifts, we then carry out a simulation analysis for each of the overlapped spectral patterns in a stepwise manner, first starting from those with two signals, then moving on to those with three signals, and so on. If any spectral pattern fits well the experimental data, one can determine the chemical shifts and J values directly. If not, one has to follow the iterative loop process (labeled by dotted lines) to simulate each of the spectral patterns. Once it is done for all, we can put all the chemical shifts and *J*_H–H_ values together to do a full spectrum analysis. In the final step, if necessary, one can manually adjust the ^1^H chemical shifts and the *J*_H–H_ values, including linewidths, to refine the simulated spectrum (Scheme 2). A high accuracy of ±0.01 ppm for the ^1^H chemical shift determination and ±0.3 Hz for the *J*_H–H_ value determinations are achievable (Appendix A). For time-saving, we here generated an abovementioned database to store the ^1^H chemical shifts and the *J*_H–H_ values of the known compounds, such as **1** and **2**. Before start working on a “new” steroidal compound, one can first check the database whether any similar spectral pattern that are available and therefore can be used directly. Aside, it is possible to make use of structural similarity to predict ^1^H chemical shifts of “new” compound out of the known compound. For example, presumably **3** and **5** are both known compounds available in the database and **7** is the new compound, one can make use of the ^1^H chemical shifts from **3** (for ring A) and those from **5** (for rings B–D), as well as associated *J*_H–H_ values (Table 2) to simulate the full spectrum for **7**. As shown (Figure 4), by properly adjusting a few chemical shifts for better curve fitting, one can easily uncover their ^1^H chemical shifts. It is highly expected that the database will grow rapidly, while more and more information to be included, which greatly facilitates the analysis.

### 2.5. Deducing Dihedral Angles and Structural Determinations

Using the Bothner-By equation [20], we calculated the dihedral angles of **1**, **2**, **3**, **4**, **5**, and **7** from three-bond *J*_H–H_ (^3^*J*_H–H_) values, respectively. These data are in good agreement with that extracted from their X-ray crystal structures (Appendix A), with only a few exceptions seen in rings A and D showing deviations greater than 10°. The discrepancies were possibly due to electronegativity in the steroidal rings A and D. At the final step of the scheme, one can make use of the ^3^*J*_H–H_ values to constitute the molecular structure of steroids or related compounds. In this work, we conducted an ab initio computer modeling simulation for **1**, **2**, **3**, **4**, **5**, and **7** while setting their dihedral angles as angular constrains. Without violating the NOE distance constraints (Appendix A), the most probable conformations resulting from the energy minimization in the simulation allow us to decipher its molecular structure accordingly. And the resulting molecular structures for compounds **1**, **2**, **3**, **4**, **5**, and **7** (Figure 5) are in good agreements with that observed by X-ray crystallography [21,22,23,24,25,26]. Based on this, we claimed that the molecular structures resolved from the ^3^*J*_H–H_ are validated.

## 3. Discussion

Despite that steroidal compounds **1** and **2** are structurally similar, however, their ^1^H NMR spectra are very different. To explain why, we devoted great efforts to analyze their ^1^H chemical shifts and associated *J*_H–H_ coupling constants, respectively. Here we reported that their *J*_H–H_ coupling constants remain basically invariant, however, their ^1^H chemical shifts deviate substantially. Due to the chemical shift deviations, it explains why their ^1^H spectra are dissimilar. As believed, the *J*_H–H_ coupling constants are universal and applicable to steroidal compounds as well as other organic compounds. Here we proposed a direct and simple approach for unraveling both *J*_H–H_ coupling constants and ^1^H chemical shifts of steroidal compounds. Making use of the *J*_H–H_ values extracted from various steroidal structures, one can easily determine their ^1^H chemical shifts without going through time-consuming 2D measurements and related data analysis.

It is anticipated that this NMR study approach will have a great impact in the field of steroidal conformational analyses and steroidal drug developments. In the application of steroid/metal ion chelation, we reported previously that ring D is responsible for the metal ion chelation and therefore its conformation is rather sensitive to the presence of metal ions, such as Mg^2+^ and Ca^2+^ [27]. By this means, one can probe steroidal conformational change due to the presence the metal ions. More conformational analyses of steroid/metal ion mixtures are currently undergoing in our lab. Apart from this, the simulation scheme we report here can be applicable for studying organic and inorganic compounds as well. As demonstrated, the *J*_H–H_ coupling constants deduced from a certain type of organic or inorganic compounds remain in principle invariant, only the relevant ^1^H chemical shifts might vary from sample to sample. By the same token, one can generate a set of ^1^H chemical shift database of a related compound. It shall be aware that the spectral patterns measured at different magnetic field strengths are rather different because the chemical shifts are field strength dependent and yet the *J*_H–H_ coupling splitting are field strength independent. Aside, the ^1^H signal patterns of organic or inorganic compounds are undoubtedly sensitive to salt concentration and solvent being used. To avoid spectral overlaps due to solvent signal, one can use deuterated methanol as D-solvent in the NMR measurements. Thus, it is advised to measure the spectra under the same condition throughout the study, including the magnetic field strength and the solvent as well.

## 4. Materials and Methods

### 4.1. Sample Preparation

The commercially available androstanolone, epiandrosterone, prednisolone, testosterone, estradiol and hydrocortisone (purity ≥ 99%) were purchased from Sigma-Aldrich (St. Louis, MO, USA) and used without further purification. Pregnenolone derivative, namely, 3β, 21-dihydroxypregna-5-en-20-one was synthesized based on the following procedure. Potassium carbonate (25 mg, 0.18 mmol) was added to a solution of 3β, 21-acetoxypregna-5-en-20-one (200 mg, 0.54 mmol) in anhydrous methanol (20 mL) and the resulting mixture stirred at room temperature for 1 h. The reaction mixture was quenched with acid resin and filtered. The residue was purified using column chromatography on silica gel (hexane/EtOAc:4/1) to afford the desired product (195 mg, 98%) as a white solid. mp 271–274 °C. (Lit. [28] 273–274.5 °C), TLC (*R*_f_ = 0.37, EtOAc/hexane = 1/1) ^1^H NMR (500 MHz, CD_3_OD): δ 5.37–5.33 (m, 1H, 6-H), 4.21 (d, *J* = 19.4 Hz, 1H, 21-H_b_), 4.15 (d, *J* = 19.4 Hz, 1H, 21-H_a_), 3.44–3.36 (m, 1H, 3-H), 2.59 (dd, *J* = 10.2 Hz, *J* = 9.2 Hz, 1H, 17-H), 2.28–2.23 (m, 1H, 4α-H), 2.26–2.20 (m, 1H, 4β-H), 2.22–2.14 (m, 1H, 16β-H), 2.06–1.98 (m, 1H, 7β-H), 1.97–1.91 (m, 1H, 12β-H), 1.91–1.85 (m, 1H, 1β-H), 1.84–1.77 (m, 1H, 2α-H), 1.77–1.70 (m, 1H, 15α-H), 1.75–1.66 (m, 1H, 16α-H), 1.70–1.63 (m, 1H, 11α-H), 1.63–1.55 (m, 1H, 7α-H), 1.59–1.47 (m, 1H, 11β-H), 1.58–1.52 (m, 1H, 2β-H), 1.55–1.45 (m, 1H, 8-H), 1.47–1.39 (m, 1H, 12α-H), 1.35–1.25 (m, 1H, 15β-H), 1.25–1.17 (m, 1H, 14-H), 1.14–1.05 (m, 1H, 1α-H), 1.03 (s, 3H, 19-CH_3_), 1.05–0.97 (m, 1H, 9-H), 0.66 (s, 3H, 18-CH_3_); ^13^C NMR (125 MHz, CD_3_OD): δ 212.1, 142.4, 122.4, 72.5, 70.3, 60.0, 58.3, 51.7, 45.7, 43.1, 39.9, 38.7, 37.9, 33.4, 33.1, 32.4, 25.8, 24.1, 22.3, 20.0, 13.9; HRMS (EI^+^): *m*/*z* calcd for C_21_H_32_O_3_: 332.2351; found: 332.2358.

### 4.2. NMR Experiments

Steroids were dried via lyophilization for 3 h to remove adsorbed moisture completely before NMR detection. Approximately 3 mg of sample was dissolved in 500 µL of anhydrous *d*_4_-methanol (Sigma-Aldrich, St. Louis, MO, USA) and analyzed using high-resolution NMR spectroscopy. For sake of consistency, all NMR measurements were performed on Bruker AV500 MHz spectrometer (Bruker BioSpin GmbH, Rheinstetten, Germany) equipped with a 5 mm z-gradient CryoProbe Prodigy BBO probe head at 298 K. All spectra were calibrated using the residual deuterated solvent signals as an internal reference (*d*_4_-methanol, ^1^H δ 4.87 ppm; ^13^C δ 49.15 ppm) and processed with Topspin 3.6 (Bruker BioSpin GmbH, Germany). For 2D experiments, ^1^H/^1^H homonuclear and ^1^H/^13^C heteronuclear chemical shift correlations were performed with the advanced version, including HSQC, HMBC, COSY, and nuclear Overhauser effect spectroscopy. The ^1^H chemical shifts and *J*_H–H_ coupling constants were deduced from the ^1^H NMR spectra via iterative full-spin analysis of steroids using the Daisy software package (Bruker BioSpin GmbH, Germany). The NMR subspectra were generated for steroidal rings of the steroidal compounds using the same software package.

### 4.3. Modelling of Steroid Structure and MD Simulation

Structures for compounds **1**, **2**, **3**, **4**, and **5** were retrieved from an online X-ray structure database while compound **7** structure was built using the GaussView software (version 4.1, Gaussian Inc., Wallingford, Connecticut, USA) [29,30]. Dihedral angles were initially set according to the calculated values from the *J* coupling constants. The OPLS-AA force field [29,30] was used for the simulation of all seven compounds. Using an OPLA force field basis model, simulation boxes containing the steroids and methanol solvents were generated. The GROMACS (2020.4) [31] package was employed as the MD engine to do the molecular dynamics simulations. All systems were analyzed comprising of the optimized steroid structures solvated with methanol inside a 5 nm cubic box. The system was initially subjected to energy minimization using the steepest descent method for 50,000 steps. The minimization step was followed by an equilibration step for 100 ps with a time-step of 2 femtoseconds at the canonical (NVT) ensemble while keeping the bonds for the steroids and methanol constrained using the LINCS algorithm. During NVT equilibration, the temperature of the system was maintained at 300 K. After NVT equilibration, it was followed by equilibration at an isothermal-isobaric (NPT) ensemble using the same parameters as in NVT equilibration while maintaining the system pressure at 1 bar using the Parrinello–Rahman isotropic coupling method. Afterward, MD simulations were then performed for all seven solvated steroid systems. After simulation, the final structure of the steroids was probed using GaussView 4.1, while setting the dihedral angles and the NOE data as angular and distance constraints, respectively.

## Data Availability

The data presented in this study are available on request from the corresponding author.

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
