# Peer review of "Resolving Entangled JH-H-Coupling Patterns for Steroidal Structure Determinations by NMR Spectroscopy"

_molecules, 2021, doi:10.3390/molecules26092643_

Round 1

Reviewer 1 Report

Wu et al. propose a procedure to assign spectral peaks and their J coupling-driven multiplet patterns in 1D NMR spectra of steroids. They show that, not surprisingly, different steroids share spectral features that can be identified by comparing different spectra. These common features can then be used to understand a spectrum of a further steroid.

This procedure is a formalization of what a spectroscopist would intuitively do when working with such spectra. Therefore, this work is helpful for researchers who work with steroids. Obviously, the approach can be adapted to other small molecules. However, new databases of relevant components of typical spectra would have to be established first.

I recommend publication of the manuscript in Molecules if the following small points can be addressed:

- There are some linguistic shortcomings; maybe a native English speaker could proof read it? (It is not severe, though)

- Page 4, lines 101-103. The authors should spell out what “differ” and “invariant” means in terms of Hz.

- This approach depends on the the B field strength since J couplings are insensitive, but chemical shifts are sensitive to it. Of course, this is considered when Hz and PPM are used to simulated spectra. Nevertheless, it should be mentioned that spectra have different patterns at different fields.

- Page 6, line 123: medium-large should be 9-6 Hz, I believe.

- Maybe, the NOEs used for structure calculation should also be tabulated in the Supporting Information

- Figure 5: It would be helpful to have the X-ray structure overlaid on the shown structures.

- The first two parts of Materials and Methods are identical to the text in the Supporting Information. Decided where it should go and only show it once.

- In there method, a TOCSY experiment is decscribed, but it is never mentioned in the text. Explain.

Reviewer 2 Report

see attached document.
